# α-Tocotrienol and Redox-Silent Analogs of Vitamin E Enhances Bortezomib Sensitivity in Solid Cancer Cells through Modulation of NFE2L1

**DOI:** 10.3390/ijms24119382

**Published:** 2023-05-27

**Authors:** Kyota Ishii, Mayuko Hido, Misaki Sakamura, Nantiga Virgona, Tomohiro Yano

**Affiliations:** 1Laboratory of Molecular Bromacology, Graduate School of Sports Health, Toyo University, Akabane City 115-8650, Japan; s4h202300016@toyo.jp; 2Department of Food and Nutritional Sciences, Faculty of Food and Nutritional Sciences, Toyo University, Itakura 374-0193, Japan; s1c112000217@toyo.jp (M.H.); s1c112000611@toyo.jp (M.S.); 3Research Institute of Life Innovation, Toyo University, Akabane City 115-8650, Japan; nantigav@yahoo.com.au

**Keywords:** vitamin E, NFE2L1, proteasome inhibitor, bortezomib

## Abstract

Proteasome inhibitors (PIs) have emerged as an attractive novel cancer therapy. However, most solid cancers are seemingly resistant to PIs. The activation of transcription factor Nuclear factor erythroid 2 related factor-1 (NFE2L1) has been characterized as a potential resistance response to protect and restore proteasome activity in cancer cells. In this study, we demonstrated that α-Tocotrienol (T3) and redox-silent analogs of vitamin E (TOS, T3E) enhanced the sensitivity of bortezomib (BTZ), a proteasome inhibitor, in solid cancers through modulation of NFE2L1. In BTZ treatment, all of T3, TOS, and T3E inhibited an increase in the protein levels of NFE2L1, the expression levels of proteasome-related proteins, as well as the recovery of proteasome activity. Moreover, the combination of one of T3, TOS, or T3E and BTZ induced a significant decrease in cell viability in solid cancer cell lines. These findings suggested that the inactivation of NFE2L1 by T3, TOS, and T3E is essential to potentiate the cytotoxic effect of the proteasome inhibitor, BTZ, in solid cancers.

## 1. Introduction

A proteasome is an enzyme complex that plays a role in degrading specific proteins and unnecessary proteins in the cells [1]. Since cancer cells have high expression of proteasome component proteins and higher proteasome activity than normal cells, proteasome may be important for cancer survival and proliferation [2,3,4,5,6]. In accordance with this, proteasome inhibitors (PIs) are known to induce effective cell death and anti-proliferation effect in some cancer cells, with some PIs in use as anticancer drugs for blood cancers like multiple myeloma [7]. Furthermore, PIs are also expected to have therapeutic effects against solid cancers, and the efficacy of PIs in several solid cancers has been investigated. However, some clinical studies have concluded that PIs are ineffectual in the treatment of solid cancers, and it is regarded that solid cancers have PI resistance [8,9,10,11,12,13].

Recently, Nuclear factor erythroid 2-related factor-1 (NFE2L1) has been identified as a protein involved in PI resistance in cancer cells [14,15,16,17]. NFE2L1 is initially present on the endoplasmic reticulum membrane (~120 kDa), and it is released from the endoplasmic reticulum to the cytoplasm by NGLY1, p97 and Hrd1 [18,19,20]. When proteasome activity is sufficient, cytoplasmic NFE2L1 is generally degraded by the proteasome. However, when the functions of proteasomes are impaired, cytoplasmic NFE2L1 is stabilized and processed into a mature form (~110 kDa) by DDI2 [15,16]. Mature NFE2L1 translocates to the nucleus and induces the expression of proteasome-related genes [21]. They then ultimately recover proteasome activity. Given these facts, the finding of inhibitors of NFE2L1 may lead to improved sensitivity to PI in solid tumors. However, the screening for inhibitors of NFE2L1 has not progressed well.

Vitamin E is a kind of phytochemical that has a long-term historical relationship to cancer. While vitamin E is well regarded as useful for cancer prevention due to its antioxidant effect, tocotrienols, a member of the vitamin E family, and redox-silent analogs of vitamin E such as α-tocopheryl succinate (TOS) and 6-O-Carboxypropyl-alpha-tocotrienol (T3E) are also known to exhibit anticancer effects via inhibition of specific cancer-related molecules such as HIF, Src, and STAT3 [22,23,24,25,26,27]. Furthermore, previous studies have reported that γ-tocotrienol and T3E modulate the expression levels of some proteasome-component proteins regulated by NFE2L1 [28,29]. Therefore, vitamin E and its redox-silent analogs may affect NFE2L1. 

In this study, we demonstrate that α-tocotrienol (T3), TOS, and T3E enhance sensitivity to bortezomib, a proteasome inhibitor, in solid cancer cells through modulating protein levels of NFE2L1.

## 2. Results

### 2.1. T3, TOS, and T3E Suppressed the Increase in Protein Levels of NFE2L1 Induced by Bortezomib Treatment

NFE2L1 is known to stabilize and translocate from cytoplasm to nucleus under proteasome inhibition [30]. First, we confirmed the protein levels of NFE2L1 in the cytoplasm and nucleus by immunoblotting. For solid cancer, we used the malignant pleural mesothelioma, H2452, due to its reported resistance to bortezomib [13,31]. As shown in Figure 1a, two bands were identified in the cytoplasmic fraction under bortezomib treatment, and the lower band was clearly identified in the nuclear fraction. Therefore, we assumed that the upper band is unprocessed NFE2L1 without transcriptional activity and the lower band is processed NFE2L1 with transcriptional activity.

Next, we used an immunoblot to evaluate the effect of TP (α-tocopherol), T3, TOS, and T3E on the protein levels of NFE2L1 under bortezomib treatment. As shown in Figure 1b, bortezomib alone demonstrated a significant increase in protein levels of unprocessed and processed NFE2L1 compared to the control group, while the combination group with bortezomib and T3, TOS, and T3E did not show this increase. Also, TOS, and T3E alone groups showed a trend toward a decrease in NFE2L1 protein level. These results suggest that T3, TOS, and T3E modulate protein levels of NFE2L1 under proteasome inhibition.

### 2.2. T3, TOS, and T3E Suppressed the Increase in Expression Levels of Proteasome-Related Proteins Induced by Bortezomib Treatment

It is known that the expression levels of proteasome-related proteins are increased by NFE2L1 under proteasome inhibition [14,32]. We also speculated that T3, TOS, and T3E might also moderate expression levels of proteasome-related proteins under proteasome inhibition. Consequently, we next evaluated the effects of TP, T3, TOS, and T3E on the expression levels of the proteasome-component proteins, PSMA7, PSMB7, and PSMC4, as well as POMP, a proteasome maturing protein, under bortezomib treatment by RT-PCR assay. As demonstrated in Figure 2a, the bortezomib alone group showed a significant increase in the mRNA expression levels of each protein compared to the control group, while the combination groups of bortezomib with T3, TOS, and T3E did not show as significant an increase as that in the bortezomib alone group. These results suggest that T3, TOS, and T3E modulate the expression levels of proteasome-related proteins under proteasome inhibition.

### 2.3. T3, TOS, and T3E Suppressed the Recovery of Proteasome Activity under and after Bortezomib Treatment

Under or following proteasome inhibition, NFE2L1 is known to recover proteasome activity [14]. We hypothesized that T3, TOS, and T3E might inhibit the recovery of proteasome activity under or after proteasome inhibition. Subsequently, we next evaluated the effects of TP, T3, TOS, and T3E on the recovery of proteasomal chymotrypsin-like activity undergoing and following bortezomib treatment. Proteasome activity was significantly reduced in all treatment groups compared to the control group (Figure 3a). After removing bortezomib from each treatment group, proteasome activity recovered to approximately 60% in the bortezomib alone group, whereas it was less than 40% in the combination groups with bortezomib and T3, TOS, T3E (Figure 3b).

Since ubiquitinated proteins are well known to accumulate under proteasome inhibition, T3, TOS, and T3E may enhance the accumulation of ubiquitinated proteins induced by PIs via inhibition of the recovery in proteasome activity. For this reason, we next evaluated the effect of the combination of bortezomib with TP, T3, TOS, and T3E on the accumulation of ubiquitinated protein by an immunoblot. As seen in Figure 3c, the combination groups with bortezomib and T3, TOS, and T3E showed a more remarkable accumulation of ubiquitinated protein compared to the bortezomib alone group. These results suggest that T3, TOS, and T3E inhibit the recovery of proteasome activity under and after proteasome inhibition.

### 2.4. T3, TOS, and T3E Enhanced Sensitivity to Bortezomib in H2452

We next evaluated the effect of bortezomib in combination with TP, T3, TOS, and T3E on cell viability using a WST-8 assay. As shown in Figure 4, the combination groups with bortezomib and T3, TOS, and T3E showed a significant decrease in cell viability compared to the control group, bortezomib alone group, and respective alone groups, suggesting that T3, TOS, and T3E may enhance bortezomib sensitivity.

### 2.5. T3, TOS, and T3E Also Enhanced the Sensitivity to Bortezomib through Protein Levels of NFE2L1 and NRF3 in Other Solid Cancer Cell Lines

Based on our results, T3, TOS, and T3E may enhance the sensitivity to bortezomib in the H2452 cell line by modulating protein levels of NFE2L1. However, it is unclear whether the same effect can be achieved in other solid cancer cell lines. Therefore, we performed an evaluation of the effects of T3, TOS, and T3E on NFE2L1 and sensitivity to bortezomib in the lung adenocarcinoma cell line, A549, and the pancreatic cancer cell line, PANC1. As a result, the bortezomib alone group showed an increase in protein levels of unprocessed and processed NFE2L1 and mRNA expression levels of proteasome-related proteins compared to the control group, while the combination groups with bortezomib and T3, TOS, and T3E did not show a similar increasing tendency as bortezomib alone group in PANC1 and A549 (Figure 5a–d). In addition, the combination groups with bortezomib and T3, TOS, and T3E demonstrated a significant decrease in cell viability compared to the control group, bortezomib alone group, and the respective alone groups in PANC1 and A549 (Figure 5e,f). These results suggest that T3, TOS, and T3E may enhance the sensitivity to bortezomib in different types of solid cancers by modulating protein levels of NFE2L1.

### 2.6. Atorvastatin Did Not Affect NFE2L1 and NRF3 under Bortezomib Treatment

Vitamin E and its derivatives are well known to have a cholesterol-lowering effect. It also has been reported that NFE2L1 is involved in the regulation of cholesterol [33]. Based on these findings, we speculated that the cholesterol-lowering effect of T3, TOS, and T3E could moderate the protein levels of NFE2L1. Therefore, finally, we evaluated the effect of atorvastatin, a cholesterol-depleting agent, on NFE2L1 under bortezomib treatment. We observed no significant differences between the bortezomib alone group and the combination group with bortezomib and atorvastatin in protein levels of unprocessed and processed NFE2L1 (Figure 6a), mRNA expression levels of PSMB7 (Figure 6b) and the accumulation of ubiquitinated proteins (Figure 6c). These results suggest that the cholesterol-lowering effect does not affect NFE2L1 under proteasome inhibition.

## 3. Discussion

In this study, we initially examined the effects of TP, T3, TOS, and T3E on NFE2L1 and proteasome homeostasis under bortezomib treatment. We observed that T3, TOS, and T3E (but not TP) suppressed the increase in protein levels of NFE2L1, as well as the expression levels of transcriptional target genes such as proteasome-component proteins (PSMA7, PSMB7, and PSMC4) and proteasome maturation proteins (POMP). Furthermore, it was observed that T3, TOS, and T3E inhibited the recovery of proteasome activity under and after bortezomib treatment and that the combination with bortezomib and T3, TOS, and T3E significantly reduced cell viability compared to them alone.

Under proteasome inhibition, NFE2L1 is known to synthesize new proteasomes by promoting the transcription of proteasome-related genes to maintain proteasome homeostasis. It has been reported that their inhibition prevented recovery of proteasome activity during proteasome inhibition and greatly enhanced the sensitivity to proteasome inhibitors in solid tumor cells such as breast cancer [14]. Since similar events to these reports were observed in the present study, T3, TOS, and T3E may also enhance sensitivity to bortezomib in solid cancer cells by targeting NFE2L1 under proteasome inhibition. This suggests that T3, TOS, and T3E may be candidate adjunctive agents for solid cancer treatment with bortezomib. On the other hand, NFE2L2 and NFE2L3, which is a transcription factor belonging to the leucine zipper family like NFE2L1, has also been reported to be involved in the recovery of proteasome activity under proteasome inhibition [5,14,34,35]. Since T3, TOS, and T3E strongly inhibited the recovery of proteasome activity, they may also affect NFE2L2 and NFE2L3. However, further studies are needed to clarify the effects of T3, TOS, and T3E on NFE2L2 and NFE2L3. We also observed the NFE2L1 inhibitory effects in T3, TOS, and T3E but not in TP. When comparing TP and T3, it is known that T3 is more readily taken up in cells than TP due to the presence of a double bond in the side chain [36,37]. Therefore, T3 may have a more immediate and full effect on NFE2L1 in comparison to TP. Additionally, TOS and T3E are derivatives that block the antioxidant group in vitamin E and are not consumed as antioxidants like TP and T3. Therefore, TOS and T3E may show stronger inhibitory effects on NFE2L1 than TP.

This study revealed that TOS and T3E, which are vitamin E derivatives with blocked antioxidant groups, also exerted inhibitory effects on NFE2L1, while atorvastatin, a cholesterol-lowering agent, did not exert inhibitory effects on NFE2L1. This suggests that T3, TOS, and T3E may exert inhibitory effects on NFE2L1 without implicating the antioxidant and cholesterol-lowering effects that have been identified in vitamin E. However, in this study, we did not confirm the effects of T3, TOS, and T3E on proteins such as NGLY, p97 and Hrd1, which are involved in the release of NFE2L1 from the endoplasmic reticulum into the cytoplasm, and additional investigations are needed to elucidate the mechanism of NFE2L1 inhibition by T3, TOS, and T3E.

DDI2 is a molecule which involve in the maturation of NFE2L1, and its inhibition suppress the function of NFE2L1 [15,16,38]. In the present study, we found that T3, TOS, and T3E suppressed the increase in protein levels of NFE2L1 under proteasome inhibition without affecting the protein level of DDI2 (Appendix A), suggesting that they are a new type of NFE2L1 inhibitor. Additionally, T3, TOS, and T3E may also affect molecules involved in NFE2L1 protein regulation, which suggests that T3, TOS, and T3E may be good tools for investigating NFE2L1 regulation.

## 4. Materials and Methods

### 4.1. Reagents

All cultures and chemicals were purchased from Nacalai Tesque (Kyoto, Japan) unless otherwise indicated. Fetal bovine serum (FBS) was purchased from Bio West (Nuaillé, France). Bortezomib (a protease inhibitor) was obtained from Wako Chemicals (Osaka, Japan). TP and TOS were purchased from Sigma Aldrich (St. Louis, MO, USA), and T3 was purchased from Tama Biochemicals (Tokyo, Japan). Antibodies for NFE2L1 (#8052), ubiquitin (#3936) and Lamin B1 (#13435) were purchased from Cell Signaling Technology (Beverly, MA, USA). DDI2 antibody (sc-514004) is purchased from Santa Cruz Biotechnology (Dallas, TX, USA).

### 4.2. α-T3E Synthesis

α-T3E was synthesized from T3 according to a previously reported procedure [39]. The purity of α-T3E was confirmed by GC–MS, 1H NMR, 13C NMR, and IR. NMR and IR spectra were consistent with the structure of α-T3E. 1H NMR (CDCl3) spectrum: 1.27 (3H, s), 1.59 (9H, s), 1.67 (3H, s), 2.00 (3H, s), 2.09 (3H, s), 2.12 (3H, s), 1.70–2.15 (16H, m), 2.57 (2H, t, J = 7.8 Hz), 2.65 (2H, t, J = 6.5 Hz), 3.68 (2H, t, J = 7.7 Hz), 4.95–5.25 (3H, m), 8.5 (1H, broad). IR (KBr) spectrum: 3200–3400 cm^−1^ (carboxylic OH) and 1710 cm^−1^ (C6=O).

### 4.3. Cell Culture

H2452, PANC1, and A549 cells were purchased for ATCC (Manassas, VA, USA). PANC1 and A549 were routinely grown in RPMI1640 supplemented with 10% FBS, 50 IU/mL penicillin, and 50 μg/mL streptomycin, and H2452 cells were routinely grown in RPMI1640 supplemented with 10% FBS, 6.5 mg/mL glucose, 1 mM sodium pyruvate, 10 mM HEPES, 50 IU/mL penicillin, and 50 μg/mL streptomycin at 37 °C in a humidified atmosphere with 5% CO_2_. Exponentially growing cells were used in experiments. Cells were plated on culture plates and cultured for 24 h to permit adherence. Cells were then cultured in RPMI1640 supplemented with 2% FBS for the indicated period, and each parameter was then examined.

### 4.4. Reagent Dissolution

TP, T3, TOS, and T3E were dissolved using ethanol. Bortezomib was dissolved with ethanol and sonication. Also, ethanol was added to the control group in each assay.

### 4.5. Cellular Fractionation

H2452 cells were cultured at a density of 1 × 10^6^ cells in a 10 cm dish for 24 h and were then treated with each agent for 12 h. After the treatment, cells were collected and fractionated into cytoplasm and nucleus fractions using NE-PER™ nuclear and cytoplasmic extract reagent kit (Thermo Fisher Scientific, Waltham, MA, USA).

### 4.6. Cell Viability

The WST-8 assay was performed to evaluate the effects of each reagent on the viability of H2452, PANC1, and A549. Cells were seeded on a 96-well plate (5 × 10^3^ cells/well), cultured for 24 h, and subsequently treated with each reagent for the indicated period as described in each figure legend. After each treatment, 10 μL of WST-8 solution was applied to each well containing 100 μL of the cell suspension and incubated at 37 °C for a further 30 min in 5% CO_2_. Color development was monitored at 450 nm using a multi-well plate reader (SUNRISE Rainbow RC-R, Tecan Japan, Kanagawa, Japan).

### 4.7. Proteasome Activity

H2452 cells were seeded on a 96-well white plate (5 × 10^3^ cells/100 μL/well), cultured for 24 h, and subsequently treated with bortezomib 50 nM or bortezomib 50 nM +TP 20 µM, T3 20 µM, TOS 20 µM, T3E 20 µM for 6 h, and another 6 h except for bortezomib. After the treatment, to assess chymotrypsin-like activity in cells, 50 µL of Proteasome-Glo™ Chymotrypsin-Like Cell-Based Assay Reagent (Promega Japan, Tokyo, Japan) was added to each well, and the plate was then incubated at room temperature for 20 min. The chymotrypsin-like activity was assessed based on estimated chemiluminescence intensity using a luminometer (Infinite M1000 PRO, TECAN Japan).

### 4.8. Isolation of mRNA and Real-Time Quantitative PCR

H2452, PANC1, and A549 cells were cultured at a density of 5 × 10^5^ cells in a 60-mm dish for 24 h and were then treated with each agent for 12 to 24 h. After the treatment, cells were collected, and total RNA was isolated using the Tissue Total RNA Extraction Mini Kit (Favorgen Biotech Corp., Ping-Tung, Taiwan). Total RNA (300 ng for each sample) was used for cDNA synthesis with the ReverTra Ace qPCR RT Kit (Toyobo, Osaka, Japan). cDNA templates were analyzed by real-time PCR using Thermal Cycler Dice Real Time System Lite (TAKARA BIO INC., Shiga, Japan) and THUNDER-BIRD™ SYBR qPCR Mix (Toyobo, Osaka, Japan), with the following program: at 95°C for 10 s followed by 40 cycles at 95 °C for 15 s and at 60 °C for 1 min. Primer sets are shown in Table 1. Gene expression data were normalized to the expression of the reference gene ribosomal protein L32.

### 4.9. Immunoblotting

H2452, PANC1, and A549 cells were cultured at a density of 5 × 10^5^ cells in a 60-mm dish for 24 h and then treated with each agent for 12 to 24 h. After the treatment, cells were harvested and lysed in ice-cold Laemmli sample buffer (Bio-Rad, Berkeley, CA, USA) containing a protease inhibitor cocktail (Nakadai Tesque) and phosphatase inhibitor (Nacalai Tesque). Cells were incubated on ice for 20 min following centrifugation at 12,000 rpm at 4 °C for 10 min. Samples were electrophoresed through a 10% or 15% SDS–polyacrylamide gel and transferred to a polyvinylidene difluoride membrane using the iBlot 2 Dry Blotting System (Thermo Fisher Scientific, Waltham, MA, USA). Membranes were blocked with Blocking One P (Nacalai Tesque) for 1 h, incubated with primary antibodies for 1 h, and then incubated with the secondary antibody for 1 h. Detection was accomplished using Chemi-Lumi One Super (Nacalai Tesque) and C-DiGit (LI-COR, Lincoln, NE, USA). A densitometric analysis of each immune band was performed using Image Studio for C-DiGit (LI-COR). Molecular sizing was conducted using Protein Ladder One Plus, Triple-color for SDS-PAGE (Nacalai Tesque). Protein concentrations were assessed using the DC Protein Assay System (Bio-Rad).

### 4.10. Statistical Analysis

Differences among groups were analyzed by a one-way ANOVA followed by the Tukey test. All statistical analyses were performed using Ekuseru Toukei software ver 8.0 (Social Survey Research Information Co., Ltd., Tokyo, Japan). Differences with *p*-values of 0.05 or less were considered to be significant. All experiments were conducted with a minimum of three samples from three independent experiments, and data were expressed as means ± SD. The number of samples in each experiment is shown in the respective figure legends.

## Figures and Tables

**Figure 1 ijms-24-09382-f001:**
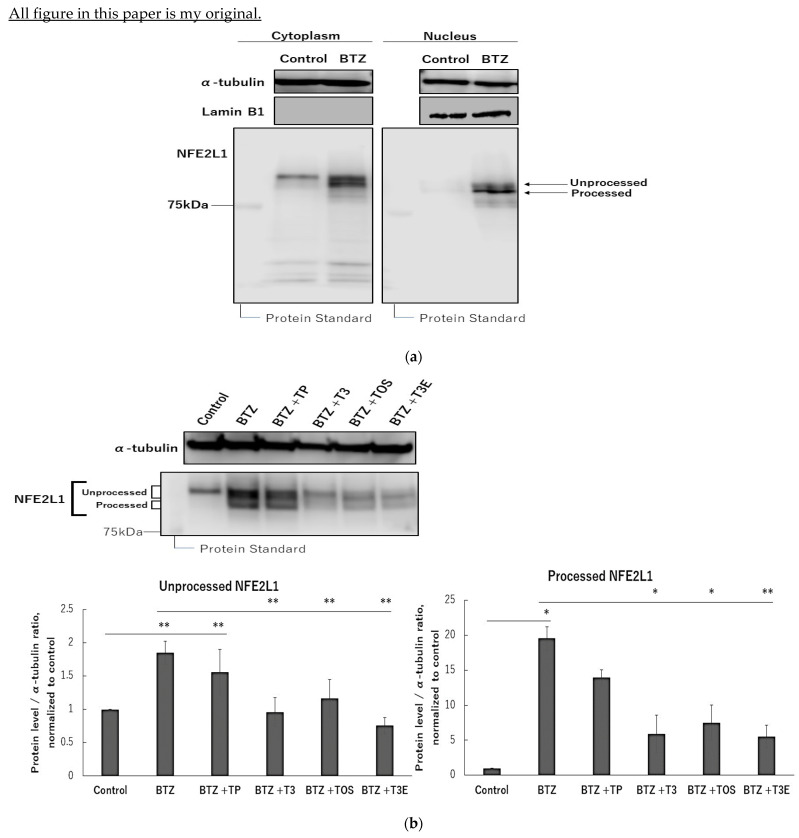
Effects of vitamin E on protein level of NFE2L1 in treatment of bortezomib. (**a**) H2452 cells were treated with bortezomib (50 nM) for 12 h and fractionated into cytoplasm and nucleus by NE-PER™ nuclear and cytoplasmic extract reagent kit. Protein levels in both fractions were assessed by immunoblotting. (**b**) H2452 cells were treated with bortezomib 50 nM or bortezomib 50 nM +TP 20 µM, T3 20 µM, TOS 20 µM, and T3E 20 µM for 12 h, and the protein levels were assessed of NFE2L1 by immunoblotting. α-Tubulin protein levels served as the loading control. A densitometric analysis was performed as described in the Materials and Methods section. Data are means ± SD, n = 3. * *p* < 0.05, ** *p* < 0.01 vs. as indicated. BTZ; bortezomib, TP; α-tocopherol, T3; α-tocotrienol, TOS; α-tocopheryl succinate, T3E; 6-O-Carboxypropyl-alpha-tocotrienol.

**Figure 2 ijms-24-09382-f002:**
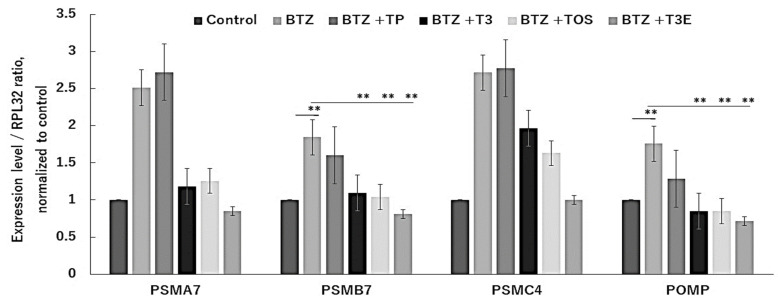
Effects of vitamin E on mRNA level of proteasome related proteins in the treatment of bortezomib. H2452 cells were treated with bortezomib 50 nM or bortezomib 50 nM +TP 20 µM, T3 20 µM, TOS 20 µM, T3E 20 µM for 12 h, and the mRNA levels of PSMA7, PSMB7, PSMC4, and POMP were assessed by real-time quantitative PCR as described in the Materials and Methods section. RPL32 mRNA levels served as the loading control. Data are means ± SD, n = 3. ** *p* < 0.01 vs. as indicated. BTZ; bortezomib, TP; α-tocopherol, T3; α-tocotrienol, TOS; α-tocopheryl succinate, T3E; 6-O-Carboxypropyl-alpha-tocotrienol.

**Figure 3 ijms-24-09382-f003:**
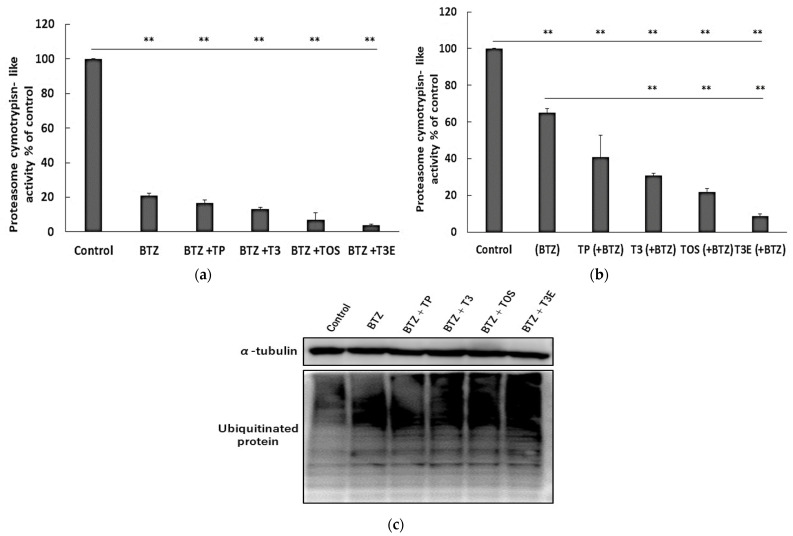
Effects of vitamin E on proteasome activities in the treatment of bortezomib. H2452 cells were treated with bortezomib 50 NM or bortezomib 50 nM +TP 20 µM, T3 20 µM, TOS 20 µM, T3E 20 µM for 6 h (**a**), and another 6 h except for bortezomib (**b**). After the treatment, chymotrypsin-like activity was assessed by a chemiluminescent method, as described in the Materials and Methods section. Data are means ± SD, n = 3. ** *p* < 0.01 vs. as indicated. (**c**) H2452 cells were treated with bortezomib 50 nM or bortezomib 50 nM + TP 20 µM, T3 20 µM, TOS 20 µM, and T3E 20 µM for 12 h. After the treatment, ubiquitinated protein levels in each sample were assessed by immunoblotting. α-Tubulin protein levels served as the loading control. Results are representative of three independent experiments. BTZ; bortezomib, TP; α-tocopherol, T3; α-tocotrienol, TOS; α-tocopheryl succinate, T3E; 6-O-Carboxypropyl-alpha-tocotrienol.

**Figure 4 ijms-24-09382-f004:**
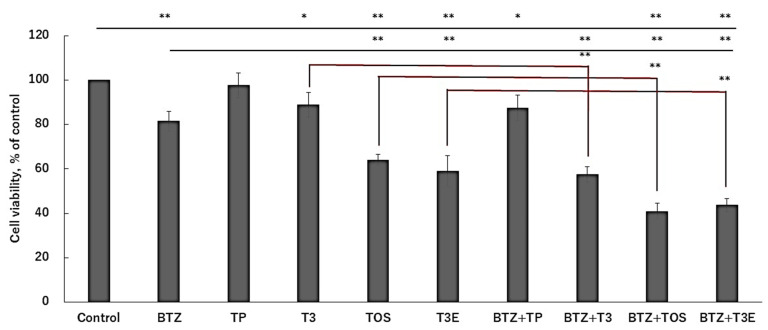
Effects of the combination of bortezomib and vitamin E on the viabilities. H2452 cells were treated with bortezomib 50 nM or bortezomib 50 nM +TP 20 µM, T3 20 µM, TOS 20 µM, T3E 20 µM for 24 h, and cell viability was evaluated by the WST-8 assay. Data are means ± SD, n = 5. * *p* < 0.05, ** *p* < 0.01 vs. as indicated. BTZ; bortezomib, TP; α-tocopherol, T3; α-tocotrienol, TOS; α-tocopheryl succinate, T3E; 6-O-Carboxypropyl-alpha-tocotrienol.

**Figure 5 ijms-24-09382-f005:**
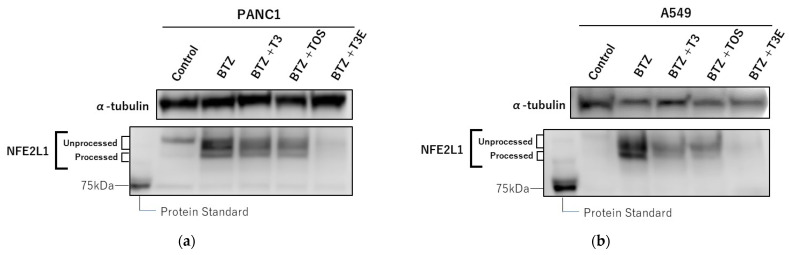
Effects of vitamin E on NFE2L1 in another solid cancer cells. PANC1 (**a**) and A549 (**b**) cells were treated with bortezomib 50 nM or bortezomib 50 nM +TP 20 µM, T3 20 µM, TOS 20 µM, T3E 20 µM for 12 h, and NFE2L1 protein levels were assessed by immunoblotting. Results are representative of three independent experiments. After PANC1 (**c**) and A549 cells (**d**) were treated with bortezomib 50 nM or bortezomib 50 nM +TP 20 µM, T3 20 µM, TOS 20 µM, T3E 20 µM for 12 h, and the mRNA levels of PSMB7 were assessed by real-time quantitative PCR as described in the Materials and Methods section. RPL32 mRNA levels served as the loading control. Data are means ± SD, n = 3. * *p* <0.05 ** *p* < 0.01 vs. the control. PANC1 (**e**) and A549 cells (**f**) were treated with bortezomib 50 nM or bortezomib 50 nM +TP 20 µM, T3 20 µM, TOS 20 µM, T3E 20 µM for 24 h. After the treatment, cell viability was evaluated by the WST-8 assy. Data are means ± SD, n = 5. * *p* < 0.05, ** *p* < 0.01 vs. as indicated. TP; α-tocopherol, T3; α-tocotrienol, TOS; α-tocopheryl succinate, T3E; 6-O-Carboxypropyl-alpha-tocotrienol.

**Figure 6 ijms-24-09382-f006:**
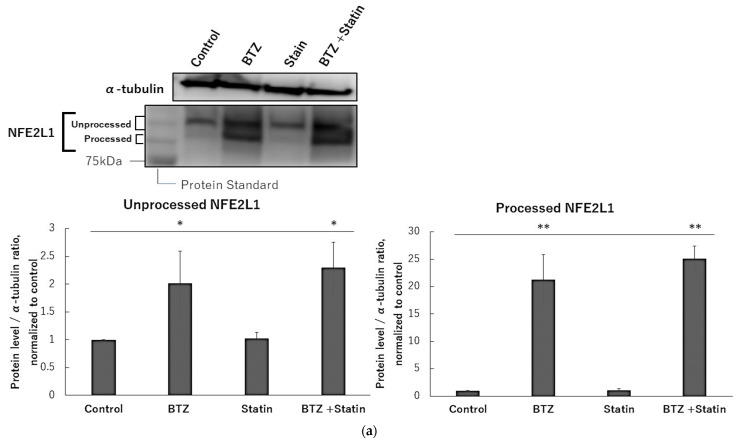
Effects of atorvastatin on NFE2L1 in treatment bortezomib. H2452 cells were treated with bortezomib 50 nM or bortezomib 50 nM +atorvastatin 10 µM for 24 h, and NFE2L1 (**a**) and ubiquitinated protein (**c**) levels were assessed by immunoblotting. α-Tubulin protein levels served as the loading control. A densitometric analysis was performed as described in the Materials and Methods section. (**b**) After H2452 cells were treated with bortezomib 50 nM or bortezomib 50 nM +atorvastatin 10 µM for 24 h, and the mRNA levels of PSMB7 were assessed by real-time quantitative PCR as described in the Materials and Methods section. RPL32 mRNA levels served as the loading control. Data are means ± SD, n = 3. * *p* < 0.05 ** *p* < 0.01 vs. the control. BTZ; bortezomib, statin; atorvastatin.

**Table 1 ijms-24-09382-t001:** List of PCR primers.

Gene Name	Primer	Sequence
60S ribosomal protein L32 (RPL32)	Forward	AACCCTGTTGTCAATGCCTC
	Reverse	CATCTCCTTCTCGGCATCA
Proteasome 20S Subunit Alpha 7 (PSMA7)	Forward	CTGTGCTTTGGATGACAACG
	Reverse	CGATGTAGCGGGTGATGTACT
Proteasome 20S Subunit Beta 4 (PSMB4)	Forward	TCAGTCCTCGGCGTTAAGTT
	Reverse	GCTTAGCACTGGCTGCTTCT
Proteasome 20S Subunit Beta 7 (PSMB7)	Forward	CGGCTGTGTCGGTGTATG
	Reverse	GCCAGTTTTCCGGACCTT
Proteasome 26S Subunit ATPase4 (PSMC4)	Forward	GGAAGACCATGTTGGCAAAG
	Reverse	AAGATGATGGCAGGTGCATT
Proteasome maturation protein (POMP)	Forward	AGGCAGTGCAGCAGGTTC
	Reverse	GGCTCTCCCATGACTTCG

## Data Availability

All relevant data are within the manuscript.

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
