# Peer review of "α-Tocotrienol and Redox-Silent Analogs of Vitamin E Enhances Bortezomib Sensitivity in Solid Cancer Cells through Modulation of NFE2L1"

_ijms, 2023, doi:10.3390/ijms24119382_

Round 1
Reviewer 1 Report
The manuscript „α-Tocotrienol and Redox-silent Analogues of Vitamin E Enhances Bortezomib Sensitivity in Solid Cancer Cells 3 Through Modulation of NRF1 and NRF3“is not written clearly and leaves several open questions due to the problems related to experimental part of the work.
Conceptually, the authors presume that BTZ treatment induces NRF1 and NRF3 and that these two transcription factors enter the nucleus (not shown), bind to the promoters of target genes (PSMA7, PSMB7, PMC4, POMP (not shown), and induce their expression (increased expression is shown, but – was it induced with NRF1/NRF2?), resulting in increased level of corresponding proteins (not shown) and proteasome function recovery (partly shown). Application vitamin E analogues has a potential for a partial reversion of the process.
The cascade of hypothesized events is not firmly experimentally confirmed.
Some parts of this manuscript ask for a significant improvement:
A) Strong statements need to be connected with appropriate references. For example:
1. Recently, Nuclear factor erythroid 2 related factor-1 (NRF1) and Nuclear factor 40 erythroid 2 related factor-3 (NRF3) have been identified as proteins involved in PI resistance in cancer cells (no ref).
2. Previous studies have reported that γ-tocotrienol and T3E modulate the expression levels of some proteasome-component proteins which are regulated by NRF1 and NRF3 (no ref).
The nomenclature is confusing. There is a gene which is officially called NRF1 (nuclear respiratory factor 1), and at the same time, NFE2L1 (official name) is also commonly called NRF1. That can be, and is highly confusing. What can reduce confusion is using the official symbols for genes/proteins (in the case of NFE2L1 it can also be TCF11), with appropriate prior explanation.
B) Some additional explanations are needed: for example, those related to the processes of NFE2L1 maturation and its cellular distribution (nuclear vs. cytoplasmic).
B) Western blots are problematic and phenomena obtained in WB need to be explained through appropriate experimental approach:
1. The authors should explain the molecular mechanisms involved in occurrence of phenomena that are manifested through proteasome dysfunction and associated with „activation of NRF1 and NRF3 into mature transcription factors “. This is even more important because WBs show pro- and active forms of proteins in question. The previous explanation of „pro “(is it glycosylated protein in the ER?) and „active “(nuclear NFE2L1?) with respect to their cellular localization is needed. The density of which band was determined?
NFE2L3: among three different forms: #1: glycosylated form in the ER; #2: non-glycosylated form found in the cytoplasm #3: N-terminally truncated form that is primarily present in the nucleus.
It is not clear what exactly is presented and in which part of the cell.
2. On Figure 1 the authors show the level of NFE2L1 in a total cellular protein fraction. Thus, they show both, cytoplasmic and nuclear cellular NFE2L1 content. The vicinity of bands (does the upper band correspond to the full-length transcript, in this setting? – or it has something with PTM?; or both?) can make a densitometry highly problematic. In my view, they should have presented the level of the NFE2L1 protein (the same applies to NRF3) separately – in the cytoplasm, and in the nucleus.
According to Novus Biologicals, the size of NFE2L3 should be below 72 kDa and only one band should be visible. I am aware of problems that may occur in these types of experiments, and for that reason I would appreciate seeing the whole image which must include the appropriate protein standard. The same count for all cropped images where protein bands are presented without a valid protein standard. This is especially relevant for the image #3, where NRF2 is presented, while specification of antibodies used for NRF2 detection is missed.
3. What was the solvent for compounds tested, including bortezomib? If anything different than the medium or water, one should add a „solvent control“ condition. A valid explanation is needed.
Finally, on Figure 1, the effect of TP is similar to that of the BTZ, and significantly different from effects induced with T3, TOS and T3E. How to explain this phenomenon?
Thank you.
I have no further comments.
Author Response
May 20, 2023
IJMS
Editor-in-Chief
Dear Sir,
Manuscript ID: ijms-2403923
Type of manuscript: Article
Title: α-Tocotrienol and Redox-silent Analogues of Vitamin E Enhances
Bortezomib Sensitivity in Solid Cancer Cells Through Modulation of NRF1 and
NRF3
We are most grateful to you and the two reviewers for the helpful comments on the original version of our manuscript. We have taken all these comments into account and submit, herewith, a revised version of our paper. In this revised version, we clearly indicated the revised points as well as additional points in red.
We have addressed the comments by reviewer 1, as indicated below, and we hope that the explanation and revisions of our work are satisfactory.
Yours sincerely
Tomohiro Yano PhD
Response to reviewer 1’s comments
- In reply to comment A) 1., we have added a reference that reports that inhibiting the process required for nuclear translocation of NRF1 improves the sensitivity of cancer to proteasome inhibitors (lines 39-45, Refs 14-21)
- In reply to comment A) ., we made some minor corrections to the text and changed the position of the references to a more understandable position (lune 56).
- In order to clearly understand the nomenclature on NRF1, We unified to NFE2L1.
- In reply to comment B) “Some additional explanations…”, we added a more detailed explanation to the introduction (Lines 39-45).
- In reply to comment B) 1. “Western blots …… The density of which band was determined”, supplementary data (Figure S1) has been added and explanations have been added (Lines 64-70).
- In reply to comments B) 1. “NFE2L3: among three different forms: #1: glycosylated form in the ER;….. and in which part of the cell.” and comment B) 2. “According to Novus Biologicals,… without a valid protein standard.”, regarding the data on NRF3, we were advised from a representative research group on NRF3 (Professor Kobayashi”s research group, Doshisha University, Kyoto, Japan) that we should spend more time performing assays with their cooperation before giving an answer. Based on this suggestion, we finally determined to completely omit the data on NRF3 in this manuscript and added additional explanation on this decision (Lines 244-248). Also, we omitted NRF3 from title in this revised manuscript (Lines 2-4).
- In reply to comment B) 2., based on the results of the fraction, we created new figures by dividing the WB results of NFE2L1 into unprocessed and processed (Figures 1,5,6).
- In reply to comment B) 2. “This is especially….NRF2 detection is misses.”, We have added NFE2L2 antibody information in Material and Method (Line 273-275),
- In reply to comment B) 2. “What was the solvent …. A valid explanation is needed,”, we have added information about the solvent and the sentence that the solvent was added to the control in Material and Method (lines 294-296).
- In reply to comment “Finally, on Figure 1, the effect ….How to explain this phenomena?”, we added a sentence explaining the difference between TP and T3, TOS or T3E in discussion (Lines 219-225).
Reviewer 2 Report
This is an interesting manuscript that describes the potential of alpha-tocotrienol, as well as non-antioxidant forms of tocochromanols. There were some minor errors that need to be corrected and a few papers the authors missed in their literature review.
Papers that should be discussed:
Tomlin FM, Gerling-Driessen UIM, Liu YC, Flynn RA, Vangala JR, Lentz CS, Clauder-Muenster S, Jakob P, Mueller WF, Ordoñez-Rueda D, Paulsen M, Matsui N, Foley D, Rafalko A, Suzuki T, Bogyo M, Steinmetz LM, Radhakrishnan SK, Bertozzi CR. Inhibition of NGLY1 Inactivates the Transcription Factor Nrf1 and Potentiates Proteasome Inhibitor Cytotoxicity. ACS Cent Sci. 2017 Nov 22;3(11):1143-1155. doi: 10.1021/acscentsci.7b00224. Epub 2017 Oct 25. PMID: 29202016; PMCID: PMC5704294.
Francois RA, Zhang A, Husain K, Wang C, Hutchinson S, Kongnyuy M, Batra SK, Coppola D, Sebti SM, Malafa MP. Vitamin E δ-tocotrienol sensitizes human pancreatic cancer cells to TRAIL-induced apoptosis through proteasome-mediated down-regulation of c-FLIPs. Cancer Cell Int. 2019 Jul 22;19:189. doi: 10.1186/s12935-019-0876-0. PMID: 31367187; PMCID: PMC6647259.
Minor comments
P2, line 66 the TP abbreviation is undefined in the text, only shown in the figure legends
Figure 1, 2 legends T3 is mis-identified as tocopherol, not tocotrienol; should be tocopheryl succinate, not tocopherol succinate
Line 94 add the word “RNA” to expression levels
Lines 115-6 “Proteasome activity was observed to significantly reduce “ change to Proteasome activity was significantly reduced…”
Lines 244 is a repetition of line 240
Largely OK, a few places the abbreviations were not correct. See comments to the authors.
Author Response
May 20, 2023
IJMS
Editor-in-Chief
Dear Sir,
Manuscript ID: ijms-2403923
Type of manuscript: Article
Title: α-Tocotrienol and Redox-silent Analogues of Vitamin E Enhances
Bortezomib Sensitivity in Solid Cancer Cells Through Modulation of NRF1 and
NRF3
We are most grateful to you and the two reviewers for the helpful comments on the original version of our manuscript. We have taken all these comments into account and submit, herewith, a revised version of our paper. In this revised version, we clearly indicated the revised points as well as additional points in red.
We have addressed the comments by reviewer 2, as indicated below, and we hope that the explanation and revisions of our work are satisfactory.
Yours sincerely
Tomohiro Yano PhD
Response to reviewer 2’s comments
- In reply to comment “Paper should be discussed:…….” We added the indicated paper:Tomlin et al., as a reference in this revised draft, and we also described the related contents in Discussion ( Ref. 18; Lines 234-237). However, with respect to other indicated paper :Francois et al., we judged that the content of this paper could not relate to the present paper, because Vitamin E does not significantly inhibit basal levels of proteasomes like proteasome inhibitors. Based on this reason, we did not cite the paper in this revised draft
- In reply to minor comments, we have completely corrected the issues you pointed out.
Round 2
Reviewer 1 Report
Most of my suggestions have been considered by authors. I believe that exclusion of NFE2L3 contributes to the clarity of this material and still leaves many relevant data.
Still:
I asked for, and I still want to see original WB for NRF2, with the protein standard included. There is nothing unusual in my request: “I am aware of problems that may occur in these types of experiments, and for that reason I would appreciate seeing the whole image which must include the appropriate protein standard. The same count for all cropped images where protein bands are presented without a valid protein standard. This is especially relevant for the image #3, where NRF2 is presented, while specification of antibodies used for NRF2 detection is missed”.
I need to see what I asked for, please: blots with standards included. For both, NFE2L1 and for NFE2L2.
Densitometries: some of them indeed are self-explanatory, so no need to adjust a nonvisible signal to “1” and then show a decrease with respect to something that already is invisible. (Figure 5; there is no signal in control A549, and if there is – it seems to be neglectable).
The primers must be listed as “forward” (and not “front”) and “reverse”.
Thank you.
No more comments.
Author Response
May 23, 2023
IJMS
Editor-in-Chief
Dear Sir,
Manuscript ID: ijms-2403923 (Rev2)
Type of manuscript: Article
Title: α-Tocotrienol and Redox-silent Analogues of Vitamin E Enhances
Bortezomib Sensitivity in Solid Cancer Cells Through Modulation of NRF1
We are most grateful to you and the two reviewers for the helpful comments on the original version of our manuscript. We have taken all these comments into account and submit, herewith, a revised version of our paper. In this revised version, we clearly indicated the revised points as well as additional points in red.
We have addressed the comments by reviewer 1, as indicated below, and we hope that the explanation and revisions of our work are satisfactory.
Yours sincerely
Tomohiro Yano PhD
Response to reviewer 1’s comments
- In reply to comment “I asked for, and I still want to see original WB for NRF2,…while specification of antibodies used for NRF2 detection is missed.”, we omitted the results of NRF2 and the sentence referring to the results of NRF2, and we added the sentence that further study is needed for NRF2 (lines 219-224).
- In reply to comment” I need to see what I asked for, please: blots with standards included. For both, NFE2L1 and for NFE2L2”, we have added a figure that shows the overall view of NRF1 with the Protein standard added in Figure 1 (Fig. 1a). Protein standard was added to the WB results for each NRF1(Figs. 1b, 5a, 5b and 6a).
- In reply to comment” Densitometries: some of them indeed are self-explanatory, …(Figure 5; there is no signal in control A549, and if there is – it seems to be neglectable)”, we excluded NRF1 protein levels in A549 and PANC1 from the numerical graphs, respectively (Fig. 5).
- In reply to comment” The primers must be listed as “forward” (and not “front”) and “reverse”, we revised the indicated points (Table1).
Round 3
Reviewer 1 Report
I thank the authors for considering all my objections and suggestions.
Although less data is now presented, the final conclusions are now adequately supported with experimental data. Although the images do not look perfect, they are convincing and informative for the readers.